# Subtyping analysis reveals new variants and accelerated evolution of *Clostridioides difficile* toxin B

Enhui Shen[1,2], Kangli Zhu[1,2], Danyang Li[1,2], Zhenrui Pan [1,2], Yun Luo[3,4], Qiao Bian[5], Liuqing He[1,2], Xiaojun Song[6], Ying Zhen[1,2], Dazhi Jin[6,7] & Liang Tao [1,2]✉

*Clostridioides difficile* toxins (TcdA and TcdB) are major exotoxins responsible for *C. difficile* infection (CDI) associated diseases. The previously reported TcdB variants showed distinct biological features, immunoactivities, and potential pathogenicity in disease progression. Here, we performed global comparisons of amino acid sequences of both TcdA and TcdB from 3,269 *C. difficile* genomes and clustered them according to the evolutionary relatedness. We found that TcdB was much diverse and could be divided into eight subtypes, of which four were first described. Further analysis indicates that the *tcdB* gene undergoes accelerated evolution to maximize diversity. By tracing TcdB subtypes back to their original isolates, we found that the distribution of TcdB subtypes was not completely aligned with the phylogeny of *C. difficile*. These findings suggest that the *tcdB* genes not only frequently mutate, but also continuously transfer and exchange among *C. difficile* strains.

[1] Key Laboratory of Structural Biology of Zhejiang Province, School of Life Sciences, Westlake University, Hangzhou, Zhejiang 310024, China. [2] Institute of Basic Medical Sciences, Westlake Institute for Advanced Study, Hangzhou, Zhejiang 310024, China. [3] Department of Microbiology, Zhejiang Provincial Center for Disease Control and Prevention, Hangzhou, Zhejiang 310051, China. [4] School of Biotechnology and Biomolecular Sciences, University of New South Wales, Sydney, NSW 2052, Australia. [5] School of Medicine, Ningbo University, Ningbo, Zhejiang 315211, China. [6] Centre of Laboratory Medicine, Zhejiang Provincial People's Hospital, People's Hospital of Hangzhou Medical College, Hangzhou, Zhejiang 310014, China. [7] School of Laboratory Medicine, Hangzhou Medical College, Hangzhou, Zhejiang 310053, China. ✉email: taoliang@westlake.edu.cn

Clostridioides difficile (formerly known as C. difficile) is a spore-forming, anaerobic, and gram-positive bacterium, that opportunistically colonizes human colon and induces diseases such as diarrhea and pseudomembranous colitis[1–3]. The symptoms of C. difficile infection (CDI) are mainly caused by two primary exotoxins, TcdA and TcdB, released from the bacterium. The molecular characterization of these toxins started in the late 1980 s, initially by cloning the toxin gene fragments. The following studies progressively mapped the chromosomal region termed pathogenicity locus (PaLoc) where the two toxin genes are located together with additional regulatory genes[4].

Both TcdA and TcdB belong to the family of large clostridial toxins (LCTs), which contain an N-terminal glucosyltransferase domain that modifies small GTPase proteins, a cysteine protease domain (CPD) that autocatalytically cleave the holotoxin in the cytosol, a combined domain for both delivery and receptor binding, and a C-terminal region consisting of series of combined repetitive oligopeptides (CROPs). These toxins enter host cells via receptor-mediated endocytosis and inactivate small GTPase proteins, leading to actin cytoskeleton disruption and cell death[5,6]. Of the two toxins, TcdB alone is able to induce a full spectrum of diseases in both animals and humans[7–9], and emerging TcdA–TcdB+ strains have been clinically isolated[10,11].

Interestingly, toxin variants of TcdB were occasionally found in nature, but no TcdA variant was ever reported to date. In 1995, EicheI-Streiber et al.[12] characterized a novel TcdB variant from C. difficile strain 1470 and named it TcdB-1470. Stabler et al.[13] later identified some potential TcdB variants by DNA microarray combined with Bayesian phylogenies; one of them was particularly intriguing because it was expressed in certain emerging hypervirulent clade 2 strains[14]. Recently, Quesada et al.[15] reported another TcdB variant in some hypervirulent clade 2 strains, which exhibits a different glycosyltransferase activity.

Owing to the fast development of sequencing techniques, numerous C. difficile genomes as well as solitary tcdA/tcdB gene sequences have been examined and submitted to the public databases such as GenBank, EMBL, and DDBJ. However, very few of these uploaded nucleotide sequences have been closely analyzed, let alone characterizing the phenotypic differences, biological activity, and hybridization properties of each toxin protein. In fact, we still lack a global view, including the diversity, evolutionary changes, and distribution of epidemic bacterial strains, of these toxin families. In this study, we retrospectively compared currently known TcdA and TcdB sequences, and further performed the subtyping analysis of TcdB.

## Results

### Sequence analysis and subtyping of TcdB.

To perform a global analysis of C. difficile toxin sequences, we obtained 3269 C. difficile genomes, including 2203 assembled genomes downloaded from NCBI Assembly, 869 raw sequenced NGS dataset from NCBI SRA and 197 newly sequenced genomes from clinical isolates (upload to NCBI database, Bio-project RPJNA591265). As a brief summary, these sequenced isolates were originated from human (n = 2322), animal (n = 137), environment (n = 265), or unknown (n = 545) sources (Supplementary Fig. 1a, Supplementary Data 1).

A total of 2868 TcdB sequences were then identified and extracted. One hundred and seventy-seven sequences were invalid or partial and they were excluded from further analysis. In total, 2691 valid TcdB sequences fell into eight major subtypes (Fig. 1a), with minimal difference between each subtype over 5.03% at amino-acid level (Table 1). TcdB1 is the largest subtype, containing 1,639 analyzed sequences (61.0% of all sequences), including toxin sequences from classic C. difficile strains such as

630 and VPI10463 (Table 1). TcdB2 group includes toxin sequences from ST1/RT027 strains, accounting for nearly one quarter of all analyzed sequences (Table 1). TcdB3 is mainly different from TcdB1 in glucosyltransferase and autoproteolytic domains (Fig. 1b); it makes up 12.4% of our analyzed sequence pool (Table 1). Interestingly, most of the strains harboring TcdB3 with available information were noted to be isolated from human sources (Supplementary Fig. 1b). Twenty-nine sequences were clustered into the TcdB4 group, TcdB4 shares considerable sequence identity to TcdB3 (~99.2%) within first 650 amino acids, but the rest part is closer to TcdB2 with the identity of ~96.4% (Fig. 1b). A previous study suggested that TcdB4 was a chimeric toxin variant evolutionary related to TcdB2 and TcdB3[15]. TcdB5-8 are newly defined TcdB variants. TcdB5 is mostly related to TcdB3 with minimal diversity of 5.03% (Table 1); the discrepancies of two subtypes mainly between amino-acid 849–973 (identity of ~85.7%), implying TcdB5 might have a different translocating efficacy compared with TcdB3. TcdB6 and TcdB7 contain variations randomly distributed through the whole sequence when compared with TcdB1 (Fig. 1b). TcdB8 is mainly different from other subtypes in the C-terminal part. All C. difficile strains containing TcdB7 were isolated from human samples; whereas the source information for isolates harboring TcdB5, 6, and 8 is largely missing (Supplementary Fig. 1b). The largest sequence divergence within TcdB is observed between subtype 4 and 8, with diversity ranging from 13.48% to 15.13% (Table 1 and Supplementary Table 1).

To test the toxicity of divergent TcdB subtypes, we expressed some of the TcdB subtypes including TcdB1, 2, 3, 4, 7, and 8, and performed toxin challenge experiments in mice by intraperitoneal injection. All tested TcdB subtypes are toxic to mice, whereas TcdB1, TcdB2, and TcdB3 were more potent compared with other subtypes (Fig. 1c).

We next investigated the geographical origin of these whole-genome-sequenced C. difficile harboring divergent TcdB variants. Interestingly, most of the C. difficile samples from North America express TcdB1 (69.9%) and TcdB2 (26.9%). In contrast, C. difficile strains isolated from East Asia mainly express TcdB1 (66.3%) or TcdB3 (29.5%). Considerable ratio of TcdB1 (48.8%), TcdB2 (35.0%), and TcdB3 (15.4%) were found in C. difficile strains from European countries. Twenty-nine samples containing TcdB4 are scattered with seven countries in Europe, North America, South America, and Asia. Besides, no TcdB3 and TcdB4 were found in samples from Australia (Fig. 1d, Supplementary Fig. 2, Supplementary Table 2).

In addition, sub-branches could be further characterized within the toxin subtypes. TcdB1 has a maximum within-subtype variation of 4.14% and consists of three clusters, which were designated as TcdB1a, TcdB1b, and TcdB1c (Supplementary Fig. 3a). Similarly, TcdB2 has a maximum within-subtype difference of 4.27% and can be further divided into two groups as TcdB2a and TcdB2b (Supplementary Fig. 3b).

### Analysis of TcdA suggests accelerated evolution of tcdB.

We also performed sequence analysis of TcdA with the same procedure. TcdA and TcdB belong to same toxin family and share similar bioactivity; and both tcdA and tcdB genes locate in the same PaLoc and are physically close to each other in the chromosome (Fig. 2a). In all, 2850 TcdA sequences were identified and extracted from the database. After removing sequences containing large truncates and insertions, 1114 sequences were considered valid and used for further analysis. Notably, truncates and insertions that disrupt the open reading frame were more frequently observed in the tcdA genes. We also performed amino-acid sequence analysis of valid TcdA sequences and found that

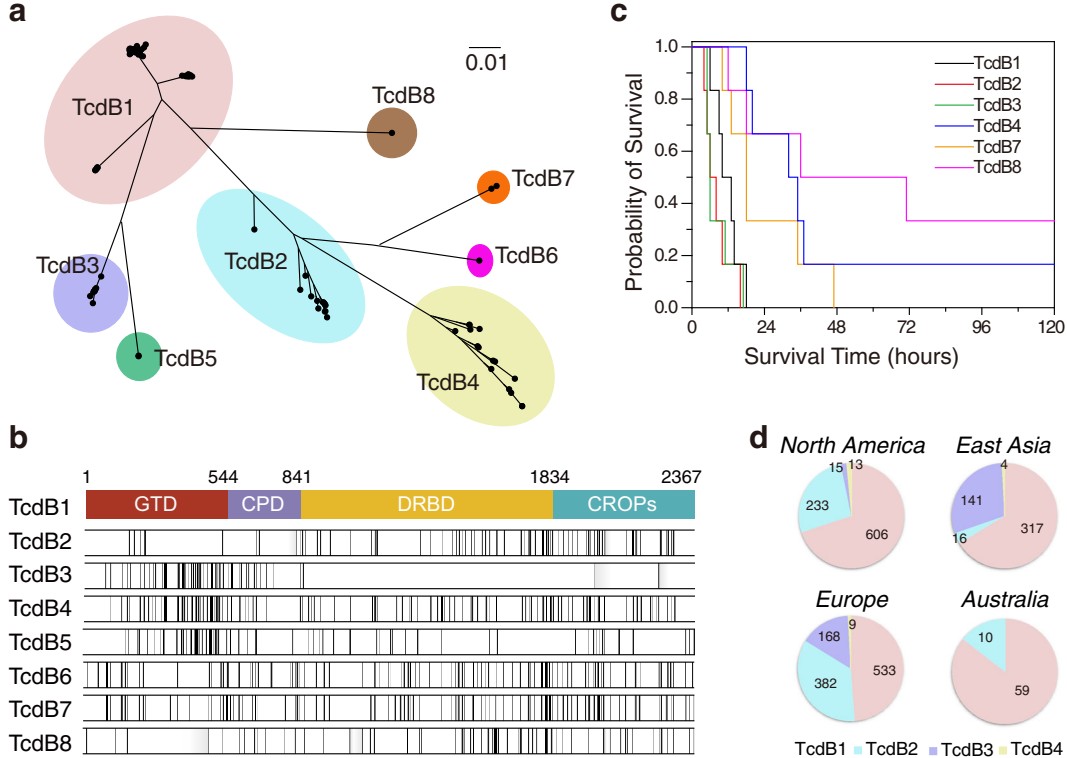

**Fig. 1 TcdB sequences are clustered into eight subtypes. a** Neighbor-joining cluster analysis of 128 unique amino-acid sequences from 2691 valid TcdB sequences. The shadows with different colors highlighted the eight TcdB subtypes with the minimal cutoff of 5.03% dissimilarity. **b** Sequence variations between TcdB1 and other TcdB subtypes. The top bar chart represents the structural arrangement of four domains (GTD, CPD, DRBD, and CROPs) in TcdB1. The black vertical lines in bar charts below mark the positions of divergent amino-acid residues in other TcdB subtypes when compared with TcdB1. **c** The time to death of C57BL/6 mice that were injected intraperitoneally with 1 μg/kg TcdB of divergent subtypes. (*n* = 6 mice) **d** Breakdown of genome-sequenced *C. difficile* strains in North America, East Asia, Europe, and Australia by TcdB subtypes 1–4.

**Table 1 Minimum between-subtype amino-acid differences among TcdB subtypes.**

| Subtype | Numbers of strains | Minimum between-subtype differences (%) | | | | | | | | Maximum within-subtype difference (%) | Representative strain |
|---|---|---|---|---|---|---|---|---|---|---|---|
| | | TcdB1 | TcdB2 | TcdB3 | TcdB4 | TcdB5 | TcdB6 | TcdB7 | TcdB8 | | |
| TcdB1 | *n* = 1639 | — | 5.33 | 5.75 | 11.15 | 5.70 | 10.06 | 10.48 | 6.72 | 4.14 | 630 |
| TcdB2 | *n* = 678 | | — | 10.10 | 5.15 | 11.24 | 6.00 | 5.75 | 8.92 | 4.27 | CD196 |
| TcdB3 | *n* = 333 | | | — | 6.38 | 5.03 | 13.44 | 13.77 | 12.13 | 0.85 | 1470 |
| TcdB4 | *n* = 29 | | | | — | 8.96 | 9.63 | 8.32 | 13.48 | 3.93 | 8864 |
| TcdB5 | *n* = 7 | | | | | — | 14.32 | 14.32 | 12.17 | 0.17 | ES130 |
| TcdB6 | *n* = 1 | | | | | | — | 5.92 | 11.33 | — | CD160 |
| TcdB7 | *n* = 3 | | | | | | | — | 11.62 | 0.17 | CD10–165 |
| TcdB8 | *n* = 1 | | | | | | | | — | — | 173070 |

TcdA were comparably conserved with an overall similarity of 97.75%, which is more conserved than some subtypes of TcdB such as TcdB1 group (similarity of 95.86%). In comparison, the minimal similarity of TcdB sequences is only 84.87% at the amino-acid level. The non-conservative residues are mainly located in the CROPs region of TcdA, whereas non-conservative residues scattered throughout the entire sequence of TcdB (Fig. 2b).

We next briefly investigated other members in the PaLoc including TcdR, TcdE, and TcdC. The amino-acid sequence analyses showed that both TcdE and TcdR are highly conserved proteins with identity positions of ~98.20% and ~96.22%, respectively. Previous studies reported that *tcdC* genes had various genotypes and were grouped into different alleles[16,17]. We clustered the *tcdC* genes following the previous method (Supplementary Fig. 4) and then investigated the association between TcdB subtype and *tcdC* group. Most of the TcdB2 associated with *tcdC* allele I and the majority of TcdB3 associated with *tcdC* allele D (Fig. 2c), implying a potential correlation between TcdB subtype and *tcdC* gene allele.

To determine the evolutionary forces that drive the observed pattern of *tcdA* and *tcdB* variation, we first partitioned nucleotide substitutions among different representative sequences into synonymous and nonsynonymous substitutions. We identified a total of 124 synonymous substitutions in *tcdA* and 714 synonymous substitutions in *tcdB*. *TcdB* harbored much more synonymous substitutions than *tcdA*, controlling for gene length. We next performed the phylogenetic analysis by maximum likelihood (PAML) among subtypes to test for the role of positive selection in *tcdA* and *tcdB* evolution using two different pairs of models including M1a/M2a and M8a/M8[18,19] (Table 2). We found that the evolution of both *tcdA* and *tcdB* was driven under

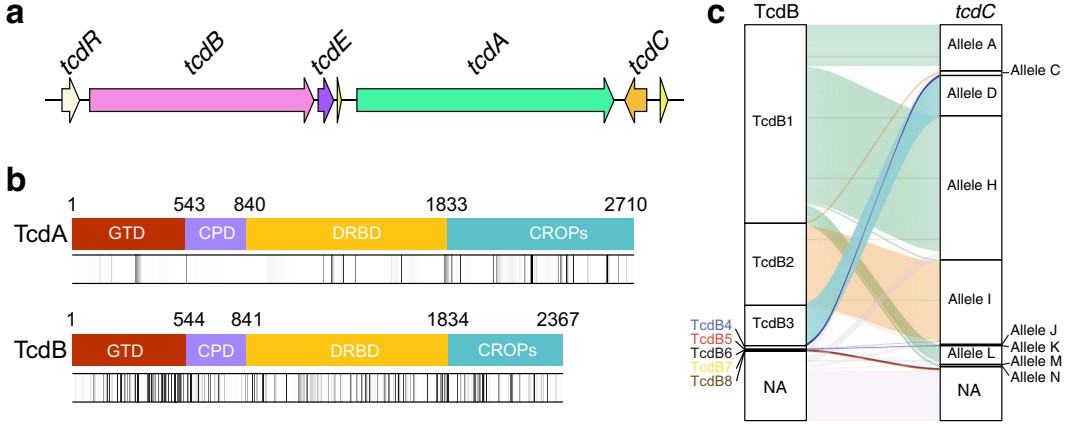

**Fig. 2 TcdB has higher sequence diversity compared with TcdA. a** The classic PaLoc structure in *C. difficile* genome referred to strain 630. The arrows represent the ORFs in the PaLoc. **b** Amino-acid sequence diversity of TcdA and TcdB. The vertical lines in the bar charts represent positions of non-conserved amino-acid residues. The gradient color (white to black) of the vertical lines indicate different conservation (100–70%) of amino-acid positions in the sequences. Positions with conservation <70% were marked as black vertical lines. **c** Riverplot graph showing associations between TcdB subtypes and *tcdC* alleles. NA means no valid sequence was identified in the *C. difficile* genome.

| Table 2 PAML analyses for the *tcdA* and *tcdB* genes. | | | | | | |
|---|---|---|---|---|---|---|
| **Models** | **Gene** | **Hypothesis (model)** | ***lnL*[a]** | ***p* value[b]** | **Hypothesis-testing** | **ω[c]** |
| M1a versus M2a | *tcdA* | Null (M1a) | −11405.514 | <0.001 | Positive selection | 5.889 |
| | | Alternative (M2a) | −11394.345 | | | |
| | *tcdB* | Null (M1a) | −18796.020 | 0.010 | Positive selection | 35.957 |
| | | Alternative (M2a) | −18791.455 | | | |
| M8a versus M8 | *tcdA* | Null (M8a) | −11394.667 | <0.001 | Positive selection | 4.869 |
| | | Alternative (M8) | −11405.514 | | | |
| | *tcdB* | Null (M8a) | −18763.111 | 0.0017 | Positive selection | 27.059 |
| | | Alternative (M8) | −18768.022 | | | |

[a]*lnL* is the log-likelihood score.
[b]Likelihood ratio test to detect positive selection, *p* < 0.05 was considered significant.
[c]Ratio of non-synonymous and synonymous substitution.

purifying selection with several positively selected sites. In addition, estimates of ω (nonsynonymous to synonymous ratio) for these positively selected sites in *tcdB* were larger than that of *tcdA*, suggesting that these sites in *tcdB* gene might evolve under stronger positive selection than in the *tcdA* gene (Table 2). This is in line with our observation that the *tcdB* gene is more diverse and accumulates more mutations (Fig. 2b).

**TcdB subtypes variably distributed in *C. difficile* strains**. Bacterial genotyping is very important for diagnosis, treatment, and epidemiological surveillance of pathogen infections and spreads. PCR ribotyping, pulsed-field gel electrophoresis, restriction-endonuclease analysis (REA), and multilocus sequence typing (MLST) are historically used in genotyping for *C. difficile*. Because of the strong Internet-accessible database support (http://pubmlst.org), MLST typing is now widely used and provides good separation for *C. difficile* isolates[20]. Therefore, we next examined the correlation between TcdB subtype distribution and *C. difficile* phylogeny based on MLST typing (Fig. 3). TcdB1 was found to be variably presented in genomes representing all *C. difficile* clades except clade 2, whereas TcdB1b was limited to the clade 3 strains and TcdB1c was limited to the clade 5 strains. TcdB2 was expressed generally by clade 2 strains but one clade 1 strain. Similarly, TcdB3 was expressed mainly by clade 4 strains and also one clade 1 strain. *C. difficile* strains of a given clade may express different TcdB subtypes, except for clade 3 (Supplementary Table 3). Moreover, we found that strains in two sequence

types (ST67 and ST41) could harbor either TcdB2 or TcdB4 (Fig. 3). It was previously proposed that TcdB4 emerged as a result of recombination events[15]. To study the potential origin of TcdB4, we conducted bootscan analysis with TcdB4 as query sequence and other TcdB subtypes as reference sequences. Given current data, we showed that TcdB4 was potentially a result of recombination among TcdB2, TcdB3, and TcdB7 (Supplementary Fig. 5).

Considering MLST typing may not best resolve the phylogeny of *C. difficile*, we also employed a genome-wide SNP based method to generate the phylogeny tree using a *Clostridium sordellii* genome as the root (Supplementary Fig. 6). By comparing the two phylogeny trees, we found that they are generally similar and particularly consistent at the clade level. The only notable difference is that ST122 strains, an outliner in the MLST tree, had been clustered to clade 1 in the genome-wide SNP tree. These findings together suggest frequent gene transfer and recombination of *tcdB*/PaLoc between *C. difficile* strains, which is consistent with the previous study[21]. Besides, TcdB6, TcdB7, and TcdB8 were only found in the outlier strains (Fig. 3), they might have independent evolutionary histories.

## Discussion
TcdB is the most important virulence factor accounting for CDI-associated diseases. Previous studies reported a few natural variants of TcdB, based on phenotypic differences, biological activity, hybridization properties, and/or relatedness to clinical

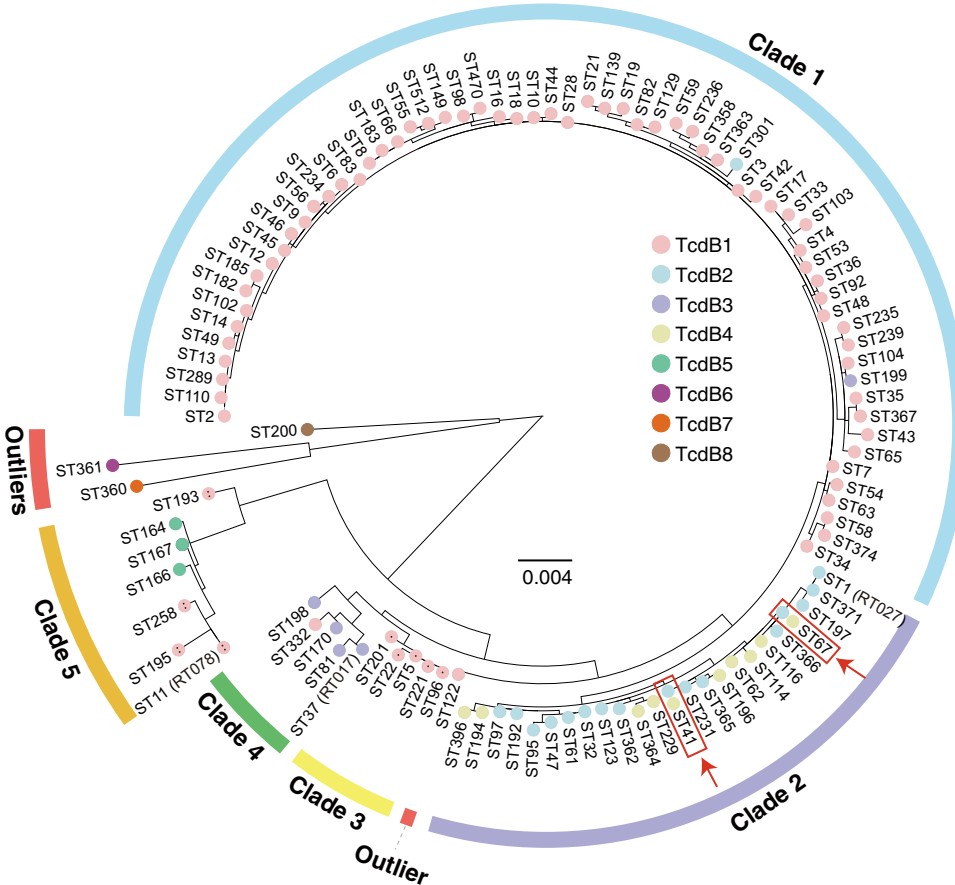

**Fig. 3 The distribution of TcdB subtypes in the phylogeny of STs.** A maximum likelihood tree was built for 110 *C. difficile* MLST types based on seven housekeeping genes (*adk*, *atpA*, *dxr*, *glyA*, *recA*, *sodA*, and *tpi*). Each color in the nodes represents one subtype of TcdB (TcdB1 in pink, TcdB2 in light blue, TcdB3 in lavender, TcdB4 in yellow, TcdB5 in green, TcdB6 in aubergine, TcdB7 in orange and TcdB8 in brown). TcdB1b and TcdB1c were denoted with one and two dots. *C. difficile* clade 1–5 were denoted by colored arcs. ST122, ST200, ST360, and ST361 are marked as outliers (red arcs). The red arrows and boxes highlight two *C. difficile* MLST types that contain either TcdB2 or TcdB4 in different isolates.

manifestation[13–15,22–24]. However, standardized subtyping for TcdB had been absent, which in part hindered characterizing the properties of these toxin variants. To establish the subtyping system for TcdB, we analyzed 2691 TcdB and 1114 TcdA sequences from 3269 *C. difficile* genomes using phylogenetic approaches. Based on these analyses, we designated a TcdB subtype that differs from the others by >5.03% at the amino-acid level. We also suggest that this defined threshold value is strict enough to distinct subtypes for bacterial toxins. As for comparisons, the minimal difference between botulinum neurotoxin subtypes is 2.6%[25], whereas the cutoff for Shiga toxin subtypes is 4.11%[26]. As expected, our subtyping system clearly divides four previously reported TcdB variants into divergent subtypes: the canonical TcdB such as $TcdB_{630}$ were divided into TcdB1, TcdB from ST1/RT027 strains belong to TcdB2, $TcdB_{1470}$ belongs to TcdB3, and $TcdB_{8864}$ belongs to TcdB4. In addition, we further identified four new subtypes (TcdB5-8). TcdB5 was identified in clade 5 *C. difficile* strains. Sequences belonging TcdB6, TcdB7, and TcdB8 were rare and only found in outlier strains. We tested some of these TcdB subtypes and found them toxic to mice by i.p. injection, suggesting all tested toxins are functionally potent. Notably, most of the sequenced *C. difficile* isolated originated from human and/or animal samples (Supplementary Fig. 1a), therefore, there could be an intrinsic sampling bias that those *C. difficile* isolates, generally belonged to the virulent *C. difficile* clade 1–4, may have higher pathogenicity[27]. The dominant occurrence of TcdB1, TcdB2, and TcdB3 in the analyzed sequences

(Supplementary Fig. 1b) may also indicate that these subtypes are more potent and closely associated with human and animal diseases. Indeed, we observed that TcdB1, TcdB2, and TcdB3 were more potent compared with other subtypes in the mice i.p. injection assays (Fig. 1c), which might in part explain the frequent occurrence of these toxin subtypes in the pathogenic *C. difficile* strains.

There is currently no standard nomination for TcdB variants, miscellaneous names referred to same toxin variants were sometimes assigned in different studies. For example, TcdB variant initially identified from NAP1/BI/027 strains was later called $TcdB_{027}$[28,29], $TcdB_{HV}$[22,30], $TcdB_{NAP1}$[15], TcdB2[31], $TcdB_{R20291}$[32,33], or TcdB-R20291[34]. In 2005, Rupnik et al.[35] proposed nomenclature of *C. difficile* toxin variants by adding the bacterial strain in which the toxin was originally found. This nomenclature was widely used for *C. difficile* toxin studies and could accurately designate each toxin to a definite amino-acid sequence. However, this nomenclature does not reflect the general properties of the toxins and could be descriptively inconvenient and confusing under certain conditions. Using subtype names would be an appropriate way to describe the protein properties of the toxin variants. Recently, Ballard et al.[31] started to use the term TcdB2 referring to the TcdB variant from ST1/RT027 strains in their studies; thus, we also named this toxin subtype TcdB2 to avoid further confusions.

Because of the accumulated amino-acid changes, TcdB from divergent subtypes may show modified biological activity,

hybridization properties, and pathogenic relatedness in varying degrees. Albeit this largely remains unclear, researchers have already begun to investigate the discrepancies among TcdB variants/subtypes, including substrate targeting, translocating effectiveness, receptor binding affinity, as well as epitopes recognized by antibodies. The classical TcdB (TcdB1) uses chondroitin sulfate proteoglycan 4[36], poliovirus receptor-like 3[37], and frizzled proteins (FZDs)[38] as cellular receptors; and can glucosylate small GTPase such as RhoA/B/C, Rac1, Cdc42, TC10, TCL, RhoG, Rap, and Ras[39–41]. Recent studies showed that TcdB2 only weakly bound to FZDs when compared with TcdB1[32,42], likely owing to discrepancies between FZD-binding sequences of TcdB1 and TcdB2[43,44]. Also, when compared with TcdB1, both TcdB3 and TcdB4 showed a drastically reduced ability to glucosylate RhoA and Cdc42[15,23], which was also supported by our sequence analyses (Fig. 1b). In addition, some studies reported that several immunoproteins including bezlotoxumab, an FDA-approved human monoclonal antibody for treating CDI[45], may have distinct neutralizing activities to TcdB1 and TcdB2[28,46] (and potentially to other TcdB subtypes), which could be vital to clinical antitoxin treatment for CDI. In some cases, TcdB subtyping together with MLST analysis could be a helpful procedure when choosing appropriate immunoproteins. Moreover, obvious sequence diversity was also observed within certain TcdB subtypes. For instance, TcdB1a and TcdB1c were found in different *C. difficile* clades and had a maximum sequence difference of 4.14%. Such within-subtype sequence variations are impressive, the potential differences in biological activity and hybridization properties between these within-subtype groups might be interesting to be further studied.

In contrast to TcdB, TcdA is much more conserved with maximum amino-acid sequence diversity of 2.25% (15.13% for TcdB). This phenomenon is intriguing, because mostly *tcdA* and *tcdB* locate in the same PaLoc and are spatially very close to each other (Fig. 2a). Previous evolutionary studies on *C. difficile* genomes showed that PaLoc in some clade 3 strains had a 9-kb insertion and in some clade, five strains had a mono-toxin arrangement, implying PaLoc in different *C. difficile* clades might have complex evolutionary history[47,48] and bi-toxin PaLoc might evolve from mono-toxin PaLoc[49]. Interestingly, we found that all the mono-toxin PaLoc had *tcdB* genes encode TcdB5-8 and vice versa, which may help to further study the evolution of *C. difficile* PaLoc. In addition, the presence of similarity of glucotransferase domain between TcdB3 and *C. sordellii* TcsL (~73%) implies the potential of inter-species recombination of the toxin genes[24]. In line with these studies, our data on synonymous substitution suggests that TcdB has a higher mutation rate or has a very different history of recombination and horizontal gene transfer[50], comparing to TcdA, which is also consistent with our observation with the genomic data. We found that TcdB was highly diverse especially when compared with TcdA, suggesting a complex and accelerated evolution of the *tcdB* gene. We also reported the TcdB subtype genes variably distributed among *C. difficile* strains and may frequently transfer, which could be alarming to the clinical prevention and treatment of CDI.

Another intriguing finding is that the distribution of TcdB subtypes is not completely aligned with the phylogeny of *C. difficile*. Because the phylogeny reflects the evolutionary relationship of bacterial strains from a historic scale, it seems that the movement of *tcdB*/PaLoc is frequent, either by horizontal gene transfer or recombination, among multiple *C. difficile* strains[21]. Clade 2 and clade 5 *C. difficile* are more genetically divergent[51] and also harbor more-divergent TcdB if individual exceptions are excluded (Fig. 3, Supplementary Table 3). TcdB4 and TcdB5 were limited to clade 2 and clade 5 strains, respectively; perhaps these are recently diverging TcdB variants. On the other hand, 29 *C.*

*difficile* samples containing TcdB4 are scattered with seven countries across Europe, America, and Asia (Supplementary Table 2), indicating the TcdB4 strains may already exist for some time that allows them to spread to multiple regions in the world. Moreover, we observed that strains from ST41 and ST67 could harbor TcdB belonged to either TcdB2 or TcdB4 (Fig. 3), indicating TcdB4 could be recently derived from TcdB2 by hybrid events. Therefore, the TcdB subtype in a *C. difficile* isolate may not be accurately defined simply by phylogenic analysis of the host bacterium.

In this study, we performed in silico subtyping analysis of *C. difficile* toxins and unveiled four new TcdB subtypes. We found that TcdB was much more diverse in amino-acid sequence than TcdA; as TcdB is particularly important for the disease, it would be interesting to find what selective pressure results in the accumulation of a large number of mutations within *tcdB* gene in the future. We also reported that the distribution of TcdB subtypes was not always correlated with the phylogeny of *C. difficile*, which could be important for clinical diagnosis and treatment of CDI. Overall, we suggest that our work would be beneficial to future studies in the toxicology of *C. difficile* toxins and epidemiology of CDI, and potentially instructive to diagnosis and therapy of the related infectious diseases.

## Methods

**Genomic data collection**. In summary, we collected 3269 genomes of *C. difficile* for analysis. Among them, 2203 assembled genomes were downloaded from NCBI. Eight hundred and sixty-nine available sequencing reads were downloaded from the recently published data sets[52]. We also generated genomic sequences from 197 newly sequenced *C. difficile* isolates (uploaded to NCBI database, PRJNA591265). Afterward, software SPAdes v3.13.1 was applied to carry out de novo assembly with standard parameters[53].

**C. difficile strain isolation and whole-genome sequencing**. A total of 197 *C. difficile* strains were isolated from anaerobic stool culture between 2011 and 2017. All stool specimens were inoculated on selective cycloserine-cefoxitin-fructose agar plates (Oxoid, Unites Kingdom) supplemented with 7% sterile defibrinated sheep blood after absolute ethanol shock treatment and incubated in an anaerobic chamber with GENbag anaer (bioMérieux, Marcy l'Étoile, France) at 37 °C for 48 h. All colonies were identified by special odor, characteristic morphology, and gram staining. Genomic DNA was extracted by using QIAamp DNA Mini Kit (Valencia, CA, USA) according to the manufacturer's instructions and later applied for whole-genome sequencing. Paired-end sequencing (2 × 150 bp) was performed by using Illumina Hiseq X-ten. Followed by removal of PCR duplicate reads with Super Deduper (github.com/dstreett/Super-Deduper); trimming of poor-quality 5′-and 3′-ends with sickle (github.com/najoshi/sickle); and removal of overlapping and adapter sequences using FLASH2 (github.com/dstreett/FLASH2). Reads shorter than 50 bp were discarded.

**Sequence analysis**. With the annotation files of 2203 genomes from NCBI, we extracted 1782 TcdB and 501 TcdA nucleotide sequences via local *Perl* scripts. These sequences were treated as queries to search against all 3269 genomes to get the best hits with BLAST v2.9.0+[54]. EMBOSS v6.6.0[55] was used to translate the nucleotide sequences into amino-acid sequences. After removing the ones with partial or invalid sequences, 2691 TcdB and 1114 TcdA amino-acid sequences were remaining for further analysis.

The whole unique amino-acid sequences of TcdA and TcdB were analyzed separately. MAFFT v7.0 was employed to achieve multiple alignments for the above sequences within default parameters[56]. Neighbor-joining trees or maximum likelihood trees were constructed by MEGA v10.0.5[57] for all unique TcdA and TcdB sequences with 1000 bootstrap simulations. FigTree v1.4.4 was used to generate the phylogenic trees.

Representative sequences of TcdB2-8 were respectively compared with TcdB1 via pairwise alignment using MAFFT v7.0. CLC Sequence Viewer v8.0.0 (CLC Bio Qiagen, Aarhus, Denmark) was adopted to visualize the variations sites in each comparison. We also extracted a unique amino-acid sequence of TcdA and TcdB from sequence pools. These sequences were aligned by MAFFT v7.0 and visualized by CLC Sequence Viewer v8.0.0 to represent diversities of TcdA and TcdB.

To study the potential recombination origin of TcdB4, we conducted bootscan analysis using the Simplot v3.5.1 software with default parameters with the TcdB4 sequence as the query.

**Production and purification of TcdB proteins**. Gene encoding different TcdB subtypes were codon optimized and synthesized by Genscript (Nanjing, China).

Genes fragments were then cloned into pHT01 vector with a 6xHis tag introduced to their C-terminus. TcdB proteins were expressed in *Bacillus subtilis* strain SL401. Bacteria were cultured at 37 °C till $OD_{600}$ reached 0.6. Expression was induced with 1 mM isopropyl-b-D-thiogalactopyranoside at 25 °C for 16 hours. Purification of His-tagged TcdB was performed by Ni-affinity chromatography and size-exclusion chromatography. All purified TcdB variants could normally induce cytopathic effect when applied to cultured cells, suggesting they are well-folded and active.

**Toxin challenge assays in mice.** C57BL/6 mice (6–8 weeks, male, specific-pathogen-free) were purchased from the Laboratory Animal Resources Center at Westlake University (Hangzhou, China). Mice were kept under specific-pathogen-free condition and given free access to normal drinking water and food during the experiments. For the toxin challenge assay, C57BL/6 mice were injected with 1 µg/kg of different TcdB subtype proteins intraperitoneally. Each group contains six mice. The animals were monitored for up to 5 days post-challenge for toxic effect and mortality, and mice were killed if they became moribund. Survival was graphed as Kaplan–Meier curves.

**Associations between TcdB subtypes and *tcdC* alleles.** The *tcdC* gene from the reference strain 630 was used to search against assembled genome and sequences of the *tcdC* gene were retrieved by local python scripts. After combing the same sequences and removing low-quality data, the remaining 63 unique sequences together with eight reference sequences obtained from the previous study (Bouvet and Popoff, 2008) were used to generate the tree with a bootstrap value of 1000 replicates. And the R package *ggalluvial* was applied to draw the riverplot for TcdB subtypes and *tcdC* alleles.

**MLST analysis.** Seven housekeeping genes (*adk, atpA, dxr, glyA, recA, sodA,* and *tpi*) in *C. difficile* were used to assign the sequence types for 3269 genomes as previously reported[20]. The python script from MLST v2.0[58] was installed in the local server to perform the MLST. In total, 110 sequence types were determined from the input *C. difficile* genomes. After extracting the sequences of seven housekeeping genes in each sequence type and multiple alignments, a maximum likelihood tree was constructed using 1000 bootstrap simulations using software MEGA.

**Whole-genome SNP typing.** In order to create the genome-wide SNP base phylogeny tree for *C. difficile*, we first filtered out the low-quality genome sequences. The remaining 3146 *C. difficile* genomes and a *C. sordellii* genome (CBA7122) were applied for the SNP detection and phylogenetic analysis by using the kSNP3 v3.1.2 software[59]. A parsimony tree was created with the core SNPs of the processed genomes. Evolview v3 was used to beautify the tree[60].

**PAML analysis for TcdA and TcdB.** Sequences were aligned by CLUSTALW using software MEGA[57], and guide trees for PAML analyses were built using RAxML v8.2.12 using the GTRGAMMA model[19]. Using PAML, we ran Codeml using four different models: M1a (neutral model) versus M2a (positive selection) and the positive selection model (M8) and its null model (M8a)[18]. The significance of differences between the two nested models was evaluated using likelihood ratio tests by calculating twice the log-likelihood of the difference following a chi-square distribution. We used DnaSP (v6.12.03)[61] to calculate the number of synonymous and nonsynonymous (replacement) substitutions in the coding regions of *tcdA* and *tcdB* genes.

**Statistics and reproducibility.** For toxin challenge assays in mice, each group contains six mice. Survival was graphed as Kaplan–Meier curves.

**Ethics statement.** The toxin challenge studies in mice were performed in strict accordance with institutional guidelines. All animal procedures reported herein were approved by the Institutional Animal Care and Use Committee at Westlake University (IACUC Protocol #19-010-TL). The procedures precluded the use of anesthesia for the toxin challenge assays. To minimize the distress and pain, the mice were monitored at least twice a day. Any animals with signs of pain or distress such as labored breathing, inability to move after gentle stimulation, disorientation, or loss of over 20% body weight were killed immediately. This method was approved by the IACUC and monitored with a qualified veterinarian.

**Reporting summary.** Further information on research design is available in the Nature Research Reporting Summary linked to this article.

## Data availability

All relevant data are available from the authors upon request. One hundred and ninety-seven newly sequenced genomes from clinical isolates have been deposited to NCBI database Bio-project RPJNA591265. Source data underlying graphs and charts shown in figures and tables are provided in Supplementary Data 1.

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

## Acknowledgements

We thank Dr. Min Dong (Boston Children's Hospital and Harvard Medical School, USA) and Dr. Yunsong Yu (Sir Run Run Shaw Hospital and Zhejiang University School of Medicine, China) for the discussions and suggestions. This study was partially supported by the National Natural Science Foundation of China (grant no. 31970129 and grant no. 31800128 to L.T., and grant no. 31900315 to Y.Z.). L.T. also acknowledges support by the Natural Science Foundation of Zhejiang Province for Distinguished Young Scholars of Zhejiang (grant no. LR20C010001). D.J. acknowledges support by the Key Research and Development Program of Zhejiang (grant no. 2015C03048). K.Z. acknowledges support by the Natural Science Foundation of Zhejiang Province (grant no. LQ20C040001).

## Author contributions

L.T. initiated and designed the project. E.S. conducted the majority of the experiments including data collection and sequence analysis. K.Z. and Y.Z. performed PAML analysis. Y.L., Q.B., X.S., and D.J. collected C. difficile isolates and performed the whole-genome sequencing for the bacterial strains. D.L. and Z.P. purified the toxin proteins and performed the mouse i.p. injection assays. D.L. and L.H. helped with data collection and analysis. E.S. and L.T. wrote the manuscript with input from all co-authors.

## Competing interests

The authors declare no conflicts of interest.
