## [Peer review file · Communications Biology]

Reviewers' comments:

Reviewer #1 (Remarks to the Author):

The study "Subtyping analysis reveals new variants and accelerated evolution of *Clostridioides difficile* toxin B" was an interesting study in the different allelic variations of *tcdB1* using a diverse dataset of genome sequences. Identifying 4 new variants was interesting, however the use of the method as a subtyping tool may not be useful due to the low frequency of the new variants. It seems the majority of *tcdB* are 1-4. Overall, the research community will benefit knowing new variants exist and the threshold in determining the variant type.

Major comments

- The major concern is the use of MLST genes to generate phylogeny when whole genome sequence data is available. In other organisms, seven housekeeping genes does not resolve actual phylogeny as house keeping genes can often undergo recombination. *C. difficile* is one particular organism that can undergo large scale recombination. I suspect the use of MLST genes in constructing the tree is causing the outliers as it appears that *tcdB6*, and *tcdB7* are the progenitors of the other *tcdB* alleles. Suggest using another species as a root.
- To improve the manuscript, the authors should perform an analysis on *tcdB* variants in relation to country of origin. It is important to know if the distribution of *tcdB* variants is found around the globe or is from a single country. For example, are all 29 samples containing *tcdB4* all from one country? Another interesting aspect that can be explored is to determine if the differences within subtypes correlate to country.
- Necessity of Figure 2b; To generate a tree, you would require a core region (common region), As both genes are different in length and different nucleotide sequences, you naturally would get two different trees. Suggest modification of the figure to just focus on *tcdA* or remove it.
- The distribution of the new *tcdB* 5-8 variants is quite low, 7, 1, 3 and 1 respectively. *tcdA*- and *tcdB*- can be obtained from *C. novyi*, *C. perfringens*, and *C. sordellii*. Since PaLoc is a region that is easily shared among *Clostridioides*, it would be interesting to use the whole regions of the PaLoc and determine the origin by analysing other species of *Clostridioides* genomes.
- Line 144; "We found that the evolution of both *tcdA* and *tcdB* were driven under purifying selection and *tcdB* gene may evolve under stronger selective pressure (Supplementary Table 2). This is in line with our observation that *tcdB* gene is more diverse and accumulates more mutations (Figure 2c)". In Supp Table 2, both genes are under purifying selection, but how can you justify *tcdB* as being under stronger selection pressure, as *tcdA* is also significant ($p < 0.05$). However, *tcdA* is shown to have less mutations than *tcdB*.
- Several figures and supplementary figures are redundant
 - o Supplementary figure 1
 - o Figure 1a

Minor Comments

Line 73-77; "We found that was highly diverse especially when compared with TcdA, suggesting a complex and accelerated evolution of the tcdB gene. We also reported the TcdB subtype genes variably distributed among *C. difficile* strains and may frequently transfer, which could be alarming to the clinical prevention and treatment of CDI" should belong to discussion rather than ending paragraph of introduction.

Line 81; 3276 genomes were mentioned, however I only counted 3270 based on the following Line 83-84 "including 2,203 assembled genomes downloaded from NCBI Assembly, 870 raw sequenced NGS dataset from NCBI SRA and 197 newly sequenced genomes from clinical isolates (upload to database, Bio-project RPJNA591265)"

Line 86; Similarly, the numbers based on origin total to 3269 "human (n=2322), animal 85 (n=137), environment (n=265), or unknown (n=545)"

Line 87-88; Country of origin total to 2658; I am assuming the major countries were listed rather than all. I would prefer not to include it the results section as this information was not discussed later, and only appeared in supplementary.

Line 162; "TcdB4 and TcdB5 were limited found in clade 2 and clade 5 strains 163 respectively; perhaps these are recent emerged TcdB variants" This statement is hard to conclude based on the figure drawn. Using housekeeping genes

Line 307- Method of whole genome sequencing was not mentioned

Line 323- The maximum likelihood tree figure contains more than TcdB1 and TcdB2, as TcdB3-8 was in the Figure.

Line 505- Expand on definition of TcdA sequence group

Line 507- The black vertical lines are in different shades in the image, grey – black, what is the significance of the grey lines

Extended supplementary data – Strain list, should also ideally include country of origin

Line 532 Unknow should be Unknown

Reviewer #2 (Remarks to the Author):

In the present work, the sequences from the two main *C. difficile* exotoxins, TcdB and TcdA, were analysed in silico in order to evaluate variability, phylogeny and forces driven variability, using a large number of *C. difficile* genomes. The findings corroborate previous ones regarding variability of the toxin-coding genes, and describe new subtypes for TcdB. The results are very descriptive, limiting their interest. Indeed, it would definitely benefit from biological studies regarding activity and virulence of the different TcdC subtypes. However, as the authors state, this work can be beneficial to future studies in toxicology of *C. difficile* and epidemiology of *C. difficile* infection.

There are however, some additional analyses that should be performed that could help explaining the higher sequence variability of TcdB in comparison to TcdA, in particular the role of recombination over mutation in generating allelic variants of TcdB. For that, authors could do a bootscan analysis, for example.

Also, the authors should mention if, among the genomes analysed, there are different arrangements of the PALOC, and whether that correlates with particular sub-types or not.

An English revision is needed.

Reviewer #3 (Remarks to the Author):

This is a well written, logically analysed piece of work which is of worth to the *C. difficile* community. The authors have analysed the toxin sequences of a large cohort of strains from both published

databases and their own collection, which encompass the global reach of *C. difficile* and the known clade groupings within them. They have noted four additional subtypes from the four known types, investigated the SNP distribution and analysed the sequence variation of these types including their clade groupings.

I have several questions regarding the conclusions of the analysis and some minor grammatical comments.

Major comments:

There is a conclusion drawn that the TcdB subtypes do not follow a pattern with MLST clades. Based on the data I don't wholly agree with this conclusion but would like clarification on some points to be convinced. For instance, do you have any statistics to demonstrate a lack of correlation? Figure 1 to me looks like the known MLST distribution of *C. difficile*, with the added detail that clade 2 TcdB appears to be diverging even further from other toxin subtypes with TcdB4. This is one of the main points of interest and should be drawn upon and emphasised. Are the TcdB4 strains later isolates (therefore more recently evolved) or is there no correlation between TcdB2/B4 emergence or divergence?

Further to this, Supplementary Figure 3 is interesting showing the three arms of TcdB1 – which clade do the more divergent arm strains (TcdB1c) belong to? If Clade 3 then that supports my interpretation of Figure 1 – where the divergent arm is like clade 3 in MLST trees, followed by clade 4 (TcdB3) and clade 5 (TcdB5) (see figure 3 from Stabler et al 2012 PLoS One for comparison).

Is it possible to generate the tree from Figure 1 with strains coloured by clade to clarify the distribution of these subtypes? Essentially this would be the inverse of Figure 3 where the MLST tree is coloured by subtype. This would either support your interpretation that there is no correlation with clade or not.

In the discussion Lines 211-221 it would be useful to refer to your data in Figure 1c – these SNPs support the literature with TcdB2 showing SNPs in the receptor binding domains (DRBD and CROPs) compared with TcdB1, and TcdB3/B4 showing SNPs in the GTD compared with TcdB1.

Brouwer et al 2016 Methods Mol Biol should be cited and discussed as this demonstrates that the PaLoc can transfer between strains of *C. difficile*, which goes some way to explaining how there is variation between strains

Minor comments:

Line 97: usually say quarter not fourth

Line 162: "limited found" is incorrect English, correct to "limited to clade..."

Line 163: these two clades are more generally genetically divergent so that would be a sound conclusion – maybe refer to their MLST divergence in this case

Lines 163-167: these are generally more discussion points and should be moved accordingly

Line 188: "belonged" should be changed to "belonging"

Line 190: "therefor" changed to "therefore"

Lines 191-194: these relate to clades 1-4 so yes, they are known to be potent, especially clades 2 and 4, perhaps refer to the literature.

Line 195: it is not good written English to say hiccup, reword to something like "There is currently no standard nomenclature for TcdB variants, miscellaneous..."

Line 211: add in "this" so, "albeit this largely..."

Lines 236-237: the meaning here is not clear. Are you referring to the phage loci insertion or something else?

Reviewer #4 (Remarks to the Author):

Shen et al. use public TcdA and TcdB sequence data in an attempt to improve on toxin subtyping to provide greater resolution to the typing schemes during *C. difficile* cases/outbreaks. The major findings from the manuscript are clear and the manuscript is written in such a way that it is easy to follow. I particularly find interesting the suggestion that genetic transfer/exchange happens with more regularity than previously thought; the implications for using non-toxic strains are evident. However, I find the manuscript light on data especially with such a hypothesis filled discussion. The authors found 8 subtypes of TcdB could be resolved using existing data and allude to the possibility that this could be a reason for the differences in toxin activity. This would be a much stronger/ more interesting finding if the authors could show how, even for some of the subtypes, the phenotypic activity of the TcdB is affected in each subtype, i.e. biological activity or receptor binding. This has been made much easier through the application of the CRISPR-cas mutation system in *C. difficile*. Additionally, I feel the findings in this manuscript could be enhanced by investigating the genetic phylogeny of the other members of the pathogenicity locus, TcdC-E. These are also important for toxin activity and subtyping techniques used by Shen et al, would provide valuable information when associating biological activity with any toxin subtyping scheme.

Minor comments

Ln 61, should read 'one of them was particularly intriguing'

Ln 72, 'retrospectively' instead of 'respectively'

Ln 163, 'recently'

Response to Reviewers (COMMSBIO-19-1865-A)

We thank reviewers for their time and support of this study. We have carried out extensive additional experiments to address reviewers' questions/suggestions. These new data largely strengthened our manuscript and we appreciate reviewers' help very much.

Reviewer #1

The study "Subtyping analysis reveals new variants and accelerated evolution of Clostridioides difficile toxin B" was an interesting study in the different allelic variations of tcdB1 using a diverse dataset of genome sequences. Identifying 4 new variants was interesting, however the use of the method as a subtyping tool may not be useful due to the low frequency of the new variants. It seems the majority of tcdB are 1-4. Overall, the research community will benefit knowing new variants exist and the threshold in determining the variant type.

Major comments

1) The major concern is the use of MLST genes to generate phylogeny when whole genome sequence data is available. In other organisms, seven housekeeping genes does not resolve actual phylogeny as house keeping genes can often undergo recombination. C. difficile is one particular organism that can under large scale recombination. I suspect the use of MLST genes in constructing the tree is causing the outliers as it appears that tcdB6, and tcdB7 are the progenitors of the other tcdB alleles. Suggest using another species as a root.

Response: We thank for the reviewer's support of this work. As the reviewer pointed out, phylogeny generated using whole-genome sequence data could usually better resolve actual phylogeny when compared with using only housekeeping genes. To address this, we employed a genome-wide SNP based method to generate the phylogeny tree (including 3,146 genome sequences with good-quality). We also added *Clostridium sordellii* CBA7122 as root as suggested. Because the newly generated phylogeny tree is very big, we put it as a supplementary figure. We then compared the phylogeny trees made by two different methods. While phylogeny generated by genome-wide SNPs provides better resolution and accuracy, both trees are generally similar and particularly consistent at the clade level. The only notable difference is that ST122 (an outlier in MLST tree, harboring TcdB1) strains have been clustered to clade 1 based on genome-wide SNPs, but this did not affect any important conclusions in our manuscript. Thus, we think that MLST based phylogeny is overall a fast and decent way for clustering *C. difficile* strains. (Line 190-199, Supplementary Figure 6)

2) To improve the manuscript, the authors should perform an analysis on tcdB variants in relation to country of origin. It is important to know if the distribution of tcdB variants is found around the globally or is from a single country. For example, are all 29 samples containing tcdB4 all from one country? Another interesting aspect that can be explored is to determine if the differences within subtypes correlate to country.

Response: We appreciate the review's advice on analyzing the geographical origin of TcdB variants. Following the suggestion, we performed the analysis on the country of origin of these whole-genome sequenced *C. difficile* strains harboring different TcdB subtypes. The geographical distribution of different TcdB subtypes varied. Most of the *C. difficile* strains from North America

express TcdB1 (69.9%) and TcdB2 (26.9%). In contrast, *C. difficile* strains isolated from East Asia mainly express TcdB1 (66.3%) or TcdB3 (29.5%). Considerable ratio of TcdB1 (48.8%), TcdB2 (35.0%), and TcdB3 (15.4%) were found in *C. difficile* strains from European countries. 29 samples containing TcdB4 are scattered with 7 countries in Europe, North America, South America, and Asia. Besides, no TcdB3 and TcdB4 were found in samples from Australia. (Line 115-123, Figure 1d, Supplementary Figure 2, Supplementary table 2)

3) *Necessity of Figure 2b: To generate a tree, you would require a core region (common region), As both genes are different in length and different nucleotide sequences, you naturally would get two both genes at both ends of the tree. Suggest modification of the figure to just focus on tcdA or remove it.*

Response: As suggested, we have removed this panel from the figure.

4) *The distribution of the new tcdB 5-8 variants is quite low, 7, 1, 3 and 1 respectively. tcdA- and tcdB-can be obtained from C. novyi, C. perfringens, and C. sordellii. Since PaLoc is a region that is easily shared among Clostridioides, it would be interesting to use the whole regions of the PaLoc and determine the origin by analysing other species of Clostridioides genomes.*

Response: PaLoc in *C. novyi*, *C. perfringens* and *C. sordellii* encode different large clostridial toxins such as Tcn α , TpeL, TcsL, and TcsH. Even though TcsL (from *C. sordellii* PaLoc) is most closely related to TcdB, it only shares an identity of ~75% with any known TcdB sequence. To our knowledge, no *tcdA* or *tcdB* genes were found in *C. novyi*, *C. perfringens*, and *C. sordellii*. Therefore, we feel that it may not be feasible to determine the origin of the new TcdB5-8 using this method.

5) *Line 144; “We found that the evolution of both tcdA and tcdB were driven under purifying selection and tcdB gene may evolve under stronger selective pressure (Supplementary Table 2). This is in line with our observation that tcdB gene is more diverse and accumulates more mutations (Figure 2c)”. In Supp Table 2, both genes are under purifying selection, but how can you justify tcdB as being under stronger selection pressure, as tcdA is also significant (p < 0.05). However, tcdA is shown to have less mutations than tcdB.*

Response: The direct estimation of selection pressure is indeed difficult. The line of evidence we used here is that the positively selected sites in the *tcdB* gene have a much higher ω (nonsynonymous to synonymous ratio) comparing to in *tcdA*. This usually suggests that these sites in the *tcdB* gene might evolve under stronger positive selection than in the *tcdA* gene. We have modified the sentences in text to better clarify this point. (Line 156-167)

6) *Several figures and supplementary figures are redundant*
o Supplementary figure 1
o Figure 1a

Response: As suggested, we have moved Figure 1a to the supplementary figure. The previous Supplementary Figure 1 has been modified with additional information on geographical distinctions of TcdB subtypes. (Supplementary Figure 1a, Supplementary Figure 2)

Minor Comments

Line 73-77; “We found that was highly diverse especially when compared with TcdA, suggesting a

complex and accelerated evolution of the tcdB gene. We also reported the TcdB subtype genes variably distributed among C. difficile strains and may frequently transfer, which could be alarming to the clinical prevention and treatment of CDI” should belong to discussion rather than ending paragraph of introduction.

Response: Following the suggestion, we have moved this part to the discussion section. (Line 286-288)

Line 81; 3276 genomes were mentioned, however I only counted 3270 based on the following Line 83-84 “including 2,203 assembled genomes downloaded from NCBI Assembly, 870 raw sequenced NGS dataset from NCBI SRA and 197 newly sequenced genomes from clinical isolates (upload to database, Bio-project RPJNA591265)”

Line 86; Similarly, the numbers based on origin total to 3269 “human (n=2322), animal 85 (n=137), environment (n=265), or unknown (n=545)”

Response: We are very sorry for the inconsistency of the numbers. We have carefully worked through the manuscript and corrected all the wrong numbers. For instance, the correct number in Line 81 should be: “we obtained 3,269 *C. difficile* genomes, including 2,203 assembled genomes downloaded from NCBI Assembly, 869 raw sequenced NGS dataset from NCBI SRA and 197 newly sequenced genomes from clinical isolates.”

Line 87-88; Country of origin total to 2658; I am assuming the major countries were listed rather than all. I would prefer not to include it the results section as this information was not discussed later, and only appeared in supplementary.

Response: As suggested, we have removed this part from the text.

Line 162; “TcdB4 and TcdB5 were limited found in clade 2 and clade 5 strains 163 respectively; perhaps these are recent emerged TcdB variants” This statement is hard to conclude based on the figure drawn. Using housekeeping genes

Response: In the newly generated phylogeny tree based on the genome-wide SNPs, we also observed that TcdB4 and TcdB5 were limited to clade 2 and clade 5 strains. We think that this may better support our statement here. (Supplementary Figure 6)

Line 307- Method of whole genome sequencing was not mentioned

Response: We are sorry that we missed that part by mistake; the method of whole-genome sequencing has been added to the methods section. (Line 369-376)

Line 323- The maximum likelihood tree figure contains more than TcdB1 and TcdB2, as TcdB3-8 was in the Figure.

Response: We have modified the text for accuracy. (Line 387-390)

Line 505- Expand on definition of TcdA sequence group,

Response: We have removed the original Figure 2b and the according figure legend has been changed.

Line 507- The black vertical lines are in different shades in the image, grey – black, what is the

significance of the grey lines

Response: The gradient color (white to black) of the vertical line indicates different conservation (100% to 70%) of an amino acid position in the sequence. All positions with conservation less than 70% were marked as black vertical lines. We have modified the figure legend with the information added. (Line 621-624)

Extended supplementary data – Strain list, should also ideally include country of origin

Response: Following the suggestion, we have added the country of origin to the Extended data 1. (Line 747-748, Extended data 1)

Line 532 Unknow should be Unknown

Response: As suggested, we have corrected the typo.

Reviewer #2

1) In the present work, the sequences from the two main C. difficile exotoxins, TcdB and TcdA, were analysed in silico in order to evaluate variability, phylogeny and forces driven variability, using a large number of C. difficile genomes. The findings corroborate previous ones regarding variability of the toxin-coding genes, and describe new subtypes for TcdB. The results are very descriptive, limiting their interest. Indeed, it would definitely benefit from biological studies regarding activity and virulence of the different TcdC subtypes. However, as the authors state, this work can be beneficial to future studies in toxicology of C. difficile and epidemiology of C. difficile infection.

Response: We strongly agree with the review that future biological studies on TcdB subtypes would be very interesting and important. On the other hand, currently we have very limited knowledge of most TcdB subtypes. To test the toxicity of divergent TcdB subtypes, we performed the toxin challenge assays in mice by intraperitoneal injection with some TcdB subtypes including TcdB1, 2, 3, 4, 7, and 8. All tested TcdB subtypes are potent to mice, suggesting they are functional toxins. Also, TcdB1, TcdB2, and TcdB3 showed higher toxicity compared to other subtypes. This result may in part explain the frequent occurrence of TcdB1, TcdB2, and TcdB3 in the pathogenic *C. difficile* strains. (Line 111-114, 230-232, Figure 1c)

2) There are however, some additional analyses that should be performed that could help explaining the higher sequence variability of TcdB in comparison to TcdA, in particular the role of recombination over mutation in generating allelic variants of TcdB. For that, authors could do a bootscan analysis, for example.

Response: We thank the reviewer for the suggestion of using additional analyses to explain a higher sequence variability of TcdB. We used the Simplot software to examine TcdB4 as an example, which was previously suggested a chimeric toxin. Using the bootscan analysis, we showed that TcdB4 was potentially a result of recombination among TcdB2, TcdB3, and TcdB7, given current data. We have added this to our manuscript. (Line 186-189, Supplementary Figure 5)

3) Also, the authors should mention if, among the genomes analysed, there are different

arrangements of the PALOC, and whether that correlates with particular sub-types or not.

Response: We searched the literature and found that there are some studies on the arrangement and evolution of *C. difficile* PaLoc before (Elliott et al. 2014, Dingle et al. 2014, Monot et al. 2015). We then analyzed the currently available genomes but did not find a new arrangement of PaLoc other than reported ones. However, we found that all the “mono-toxin PaLoc” (Monot et al. 2015) had *tcdB* genes encode TcdB5-8 and *vice versa*, which may help to further study the evolution of PaLoc. We have added it to the discussion section. (Line 273-279)

4) An English revision is needed.

Response: As suggested, an English revision has been applied.

Reviewer #3

This is a well written, logically analysed piece of work which is of worth to the C. difficile community. The authors have analysed the toxin sequences of a large cohort of strains from both published databases and their own collection, which encompass the global reach of C. difficile and the known clade groupings within them. They have noted four additional subtypes from the four known types, investigated the SNP distribution and analysed the sequence variation of these types including their clade groupings.

I have several questions regarding the conclusions of the analysis and some minor grammatical comments.

Major comments:

1) There is a conclusion drawn that the TcdB subtypes do not follow a pattern with MLST clades. Based on the data I don't wholly agree with this conclusion but would like clarification on some points to be convinced. For instance, do you have any statistics to demonstrate a lack of correlation? Figure 1 to me looks like the known MLST distribution of C. difficile, with the added detail that clade 2 TcdB appears to be diverging even further from other toxin subtypes with TcdB4. This is one of the main points of interest and should be drawn upon and emphasised. Are the TcdB4 strains later isolates (therefore more recently evolved) or is there no correlation between TcdB2/B4 emergence or divergence?

Response: We are grateful for the review's support of this work. We did not mean that there's no correlation between the TcdB subtype and MLST clade. We suggested that the pattern was not completely followed, because one *C. difficile* clade could harbor more than one TcdB subtype and one toxin subtype could be occasionally observed in *C. difficile* from different clades. We are sorry for any potential confusion here and we have modified the text to mitigate confusion. We have added a table of the distribution of TcdB subtypes in *C. difficile* clades, demonstrating that a given *C. difficile* clade could harbor multiple TcdB subtypes in a clearer way (Line 177-185, Supplementary Table 3).

We agree with the reviewer that clade 2 and clade 5 *C. difficile* strains harbor more divergent TcdB if individual exceptions are excluded, and we have added this into the discussion. (Line 293-295)

TcdB4 is a newly reported variant and was previously proposed as a chimeric toxin (Quesada-Gomez et al. 2016). TcdB4 is limited to clade 2 *C. difficile* strains, thus we postulate that TcdB4 might be an emerging variant compared to TcdB2. We performed a bootscan analysis and showed that TcdB4 was potentially a result of recombination among TcdB2, TcdB3, and TcdB7. On the other hand, 29 *C. difficile* samples containing TcdB4 are scattered with 7 countries across Europe, America, and Asia, indicating the TcdB4 strains may already exist for some time that allows them to spread to multiple regions in the world. Because genome-wide SNPs could better resolve phylogeny compared to MLST, we also employed a genome-wide SNPs based method to generate the phylogeny tree (Supplementary Figure 7). While phylogeny generated by genome-wide SNPs provide better resolution, both trees are consistent at the clade level. We tried to use this phylogeny tree to study TcdB2 and TcdB4 in clade 2 *C. difficile*. However, the exact correlation between TcdB2/B4 emergence/divergence was not clear based on the current phylogeny trees. To answer this question, we think that further studies are needed with additional evidence. (Line 184-197, 295-302, Supplementary Figure 5, Supplementary Figure 6)

2) Further to this, Supplementary Figure 3 is interesting showing the three arms of TcdB1 – which clade do the more divergent arm strains (TcdB1c) belong to? If Clade 3 then that supports my interpretation of Figure 1 – where the divergent arm is like clade 3 in MLST trees, followed by clade 4 (TcdB3) and clade 5 (TcdB5) (see figure 3 from Stabler et al 2012 PLoS One for comparison).

Response: Following the suggestion, we labeled the TcdB1b and TcdB1c in both Figure 3 and Supplementary Figure 6. As a result, strains contain TcdB1c belonged to clade 5 and strains contain TcdB1b belonged to clade 3. However, the exact correlation between TcdB subtype and *C. difficile* clades seem to be more complicated, possibly because the *tcdB* genes might evolve (by both mutagenesis and recombination) faster than the housekeeping genes in *C. difficile*. (Line 178-180, Figure 3, Supplementary Figure 6)

3) Is it possible to generate the tree from Figure 1 with strains coloured by clade to clarify the distribution of these subtypes? Essentially this would be the inverse of Figure 3 where the MLST tree is coloured by subtype. This would either support your interpretation that there is no correlation with clade or not.

Response: We tried to generate a tree from Figure 1 with strains colored by clade but found it illegible. Instead, we generated a table showing the clade distribution of TcdB. This data might reflect some correlation but also inconsistency between the TcdB subtype and *C. difficile* clade in a better way. (Supplementary Table 3)

4) In the discussion Lines 211-221 it would be useful to refer to your data in Figure 1c – these SNPs support the literature with TcdB2 showing SNPs in the receptor binding domains (DRBD and CROPs) compared with TcdB1, and TcdB3/B4 showing SNPs in the GTD compared with TcdB1.

Response: Following the suggestion, we have referred to our data in Figure 1b (previous Figure 1c). (Line 258)

5) Brouwer et al 2016 Methods Mol Biol should be cited and discussed as this demonstrates that the PaLoc can transfer between strains of *C. difficile*, which goes some way to explaining how there is variation between strains

Response: As suggested, we have included this paper as a reference to better explain how there is variation between strains. (Line 197, 293)

Minor comments:

Line 97: usually say quarter not fourth

Line 162: "limited found" is incorrect English, correct to "limited to clade..."

Line 188: "belonged" should be changed to "belonging"

Line 190: "therefor" changed to "therefore"

Line 195: it is not good written English to say hiccup, reword to something like "There is currently no standard nomenclature for TcdB variants, miscellaneous..."

Line 211: add in "this" so, "albeit this largely..."

Response: We thank the reviewer for pointing out all these grammar mistakes/typos. We have corrected them as suggested.

Lines 191-194: these relate to clades 1-4 so yes, they are known to be potent, especially clades 2 and 4, perhaps refer to the literature.

Response: As suggested, we have modified the sentence and referred to a reference (Knight et al. 2015). Moreover, we have performed the toxin challenge assays in mice by intraperitoneal injection with different TcdB subtypes. We observed that TcdB1, TcdB2, and TcdB3 showed higher toxicity compared to other subtypes. This result may also in part explain the frequent occurrence of TcdB1, TcdB2, and TcdB3 in the pathogenic *C. difficile* strains. (Line 223-232, Figure 1c)

Line 163: these two clades are more generally genetically divergent so that would be a sound conclusion – maybe refer to their MLST divergence in this case

Lines 163-167: these are generally more discussion points and should be moved accordingly

Response: As suggested, we have modified our writings here and pointed out that clade 2 and clade 5 *C. difficile* are more genetically divergent. Also, these sentences have been moved to the discussion section. (Line 293-297)

Lines 236-237: the meaning here is not clear. Are you referring to the phage loci insertion or something else?

Response: Previous studies showed that PaLoc in some clade 3 strains had the 9-kb insertion and PaLoc in some clade 5 strains showed a mono-toxin arrangement. We have modified our writings here for clearance. (Line 273-276)

Reviewer #4

I) Shen et al. use public TcdA and TcdB sequence data in an attempt to improve on toxin subtyping to provide greater resolution to the typing schemes during C. difficile cases/outbreaks. The major findings from the manuscript are clear and the manuscript is written in such a way that it is easy to follow. I particularly find interesting the suggestion that genetic transfer/exchange happens with more regularity than previously thought; the implications for using non-toxic strains are evident. However, I find the manuscript light on data especially with such a hypothesis filled discussion. The authors found 8 subtypes of TcdB could be resolved using existing data and allude to the possibility

that this could be a reason for the differences in toxin activity. This would be a much stronger/ more interesting finding if the authors could show how, even for some of the subtypes, the phenotypic activity of the TcdB is affected in each subtype, i.e. biological activity or receptor binding. This has been made much easier through the application of the CRISPR-cas mutation system in C. difficile.

Response: We are grateful for the review's support of our work. We totally agree with the review that further studies showing bioactivity differences among divergent TcdB subtypes will be very interesting. On the other hand, we only have very limited knowledge of most TcdB subtypes at this point. To test the toxicity of divergent TcdB subtypes, we performed the toxin challenge assays in mice by intraperitoneal injection with some of the TcdB subtypes including TcdB1, 2, 3, 4, 7, and 8. All tested TcdB subtypes are potent to mice, suggesting they are functional toxins. Also, TcdB1, TcdB2, and TcdB3 showed higher toxicity compared to other subtypes. This result may in part explain the frequent occurrence of TcdB1, TcdB2, and TcdB3 in the pathogenic *C. difficile* strains. (Line 111-114, 230-232, Figure 1c)

2) Additionally, I feel the findings in this manuscript could be enhanced by investigating the genetic phylogeny of the other members of the pathogenicity locus, TcdC-E. These are also important for toxin activity and subtyping techniques used by Shen et al, would provide valuable information when associating biological activity with any toxin subtyping scheme.

Response: Following the suggestion, we also investigated the phylogeny of the other members in the PaLoc including TcdC, TcdR, and TcdE. The amino acid sequence analyses showed that both TcdE and TcdR are highly conserved proteins with identity positions of ~98.20% and ~96.22% respectively. Previous studies reported that *tcdC* genes had various genotypes and were grouped as different alleles (Curry et al., 2007, Bouvet and Popoff, 2008). We clustered the *tcdC* genes following the previous method and then investigated the association between the TcdB subtype and the *tcdC* group. Most of the TcdB2 associated with *tcdC* allele I and the majority of TcdB3 associated with *tcdC* allele D, implying a potential correlation between TcdB subtype and *tcdC* gene allele. (Line 145-153, Figure 2c. Supplementary Figure 4)

Minor comments

Ln 61, should read 'one of them was particularly intriguing'

Ln 72, 'retrospectively' instead of 'respectively'

Ln 163, 'recently'

Response: Following the suggestions, we have corrected these mistakes.

REVIEWERS' COMMENTS:

Reviewer #1 (Remarks to the Author):

Authors have addressed the concerns previously brought up. It is now a much improved study that adds to the research community.

Reviewer #2 (Remarks to the Author):

The authors have replied to all my questions. I have no further comments.

Reviewer #3 (Remarks to the Author):

Thank you to the authors for addressing all points raised in my initial review. The alterations to the text, figures and inclusion of supplementary table 3 help to clarify the points raised. The inclusion and discussion of the effect of toxins in mice helped to elaborate on their data. Only one minor revision - Line 185 should be "proposed" not "purposed"

Reviewer #4 (Remarks to the Author):

The authors have made progress in their revision of the paper with several interesting findings, such as the association of tcdC allele type and tcdB variant.

The authors investigated the differential toxicity of some of the tcdB variants using an in vivo model, which shows potentially interesting findings. However, it is not clear if the authors attempted to determine if these heterologous expressed proteins folded into wildtype structures. Without this the authors cannot presume that any associated toxicity was due to tcdB variant.

Additionally, the methodology for the animals studies is missing a few key elements, such as the number of animals used in each group and how the animals were housed. The number of animals in each group is mentioned in the figure legend but nowhere else in the manuscript.

Response to the Reviewer (COMMSBIO-19-1865-B)

Reviewer #4

The authors have made progress in their revision of the paper with several interesting findings, such as the association of tcdC allele type and tcdB variant.

The authors investigated the differential toxicity of some of the tcdB variants using an in vivo model, which shows potentially interesting findings. However, it is not clear if the authors attempted to determine if these heterologous expressed proteins folded into wildtype structures. Without this the authors cannot presume that any associated toxicity was due to tcdB variant.

Response: We thank the reviewer's comments. We have validated the activities of the purified TcdB variants by testing their ability to induce cytopathic effects in the cultured cells. As expected, these TcdB proteins could normally cause cell rounding, suggesting they are well-folded and functionally active. (Line 369-370)

Additionally, the methodology for the animals studies is missing a few key elements, such as the number of animals used in each group and how the animals were housed. The number of animals in each group is mentioned in the figure legend but nowhere else in the manuscript.

Response: Mice were kept under specific-pathogen-free condition and given free access to normal drinking water and food during the experiments. For the toxin challenge assay, each group contains 6 mice. We have added these sentences into the Methods section as suggested. (Line 376-379)